# *CaWRKY30* Positively Regulates Pepper Immunity by Targeting *CaWRKY40* against *Ralstonia solanacearum* Inoculation through Modulating Defense-Related Genes

**DOI:** 10.3390/ijms222112091

**Published:** 2021-11-08

**Authors:** Ansar Hussain, Muhammad Ifnan Khan, Mohammed Albaqami, Shahzadi Mahpara, Ijaz Rasool Noorka, Mohamed A. A. Ahmed, Bandar S. Aljuaid, Ahmed M. El-Shehawi, Zhiqin Liu, Shahid Farooq, Ali Tan Kee Zuan

**Affiliations:** 1Department of Plant Breeding and Genetics, Ghazi University, Dera Ghazi Khan 32200, Pakistan; ahtraggar@yahoo.com (A.H.); mifnan@yahoo.com (M.I.K.); smahpara@gudgk.edu.pk (S.M.); hod.pbg@gudgk.edu.pk (I.R.N.); 2Department of Biology, Faculty of Applied Science, Umm Al-Qura University, Makkah 21955, Saudi Arabia; mmbaqami@uqu.edu.sa; 3Plant Production Department (Horticulture—Medicinal and Aromatic Plants), Faculty of Agriculture (Saba Basha), Alexandria University, Alexandria 21531, Egypt; mohamed_marie2009@yahoo.com; 4Department of Biotechnology, College of Science, Taif University, P.O. Box 11099, Taif 21944, Saudi Arabia; research@bssalj.com (B.S.A.); elshehawi@hotmail.com (A.M.E.-S.); 5College of Crop Sciences, Fujian Agriculture and Forestry University, Fuzhou 350001, China; 6Department of Plant Protection, Faculty of Agriculture, Harran University, Şanlıurfa 63050, Turkey; csfa2006@gmail.com; 7Department of Land Management, Faculty of Agriculture, Universiti Putra Malaysia, Serdang 43400, Malaysia

**Keywords:** chili pepper (*Capsicum annuum*), *Ralstonia solanacearum*, CaWRKY30, WRKY TFs

## Abstract

The WRKY transcription factors (TFs) network is composed of WRKY TFs’ subset, which performs a critical role in immunity regulation of plants. However, functions of WRKY TFs’ network remain unclear, particularly in non-model plants such as pepper (*Capsicum annuum* L.). This study functionally characterized CaWRKY30—a member of group III Pepper WRKY protein—for immunity of pepper against *Ralstonia solanacearum* infection. The *CaWRKY30* was detected in nucleus, and its transcriptional expression levels were significantly upregulated by *R. solanacearum* inoculation (RSI), and foliar application ethylene (ET), abscisic acid (ABA), and salicylic acid (SA). Virus induced gene silencing (VIGS) of *CaWRKY30* amplified pepper’s vulnerability to RSI. Additionally, the silencing of *CaWRKY30* by VIGS compromised HR-like cell death triggered by RSI and downregulated defense-associated marker genes, like *CaPR1*, *CaNPR1*, *CaDEF1*, *CaABR1*, *CaHIR1*, and *CaWRKY40*. Conversely, transient over-expression of *CaWRKY30* in pepper leaves instigated HR-like cell death and upregulated defense-related maker genes. Furthermore, transient over-expression of *CaWRKY30* upregulated transcriptional levels of *CaWRKY6*, *CaWRKY22*, *CaWRKY27*, and *CaWRKY40*. On the other hand, transient over-expression of *CaWRKY6*, *CaWRKY22*, *CaWRKY27*, and *CaWRKY40* upregulated transcriptional expression levels of *CaWRKY30*. The results recommend that newly characterized *CaWRKY30* positively regulates pepper’s immunity against *Ralstonia* attack, which is governed by synergistically mediated signaling by phytohormones like ET, ABA, and SA, and transcriptionally assimilating into WRKY TFs networks, consisting of *CaWRKY6*, *CaWRKY22*, *CaWRKY27*, and *CaWRKY40*. Collectively, our data will facilitate to explicate the underlying mechanism of crosstalk between pepper’s immunity and response to RSI.

## 1. Introduction

Plants frequently get exposed to various biotic and abiotic stresses during their life span due to their immobile nature [1,2]. Plants have developed several defense mechanisms in response to repeated selection pressure from major ecological and environmental constraints. Different transcriptional factors (TFs) interconnect to develop sophisticated transcriptional networks for regulating these classic defense mechanisms at a transcriptional level [3,4]. Defense reactions to various stresses are properly synchronized and controlled as they are critical for plants in the form of energy expenditure and development [5]. The defense mechanisms might vary in different plant species because of varied environmental circumstances affecting their acclimation [6,7,8]. Hence, defense system employed by model plants cannot be fully implied to non-model plants. Therefore, defense systems of plants have been a prime focus during past decades; however, these studies were mostly focused on model plants, i.e., *Arabidopsis thaliana* and rice (*Oryza sativa* L.). Nevertheless, functional synchronization of these transcriptional complexes to regulate plants’ response to various stresses has been poorly investigated, mainly in non-model plants.

The WRKY proteins comprise of the largest TFs’ family in plants. The WRKY TFs contain one or two WRKY domains and commonly WRKYGQK sequence at the N-terminus followed by C2HC or C2H2 zinc finger motif [4,9,10]. The WRKY proteins are phylogenetically divided into three main groups (i.e., group I–III) based on number of WRKY domains and structure of zinc-finger motif. The group II of WRKY TFs is again subdivided into five sub-groups (IIa, IIb, IIc, IId, and IIe) [11,12,13]. The WRKY members mainly bind with W-boxes [TTGAC(C/T)] found in promoter regions of various target genes by inserting an exclusive wedge vertically into DNA’s major groove [14] by WRKY GQK motif on the second b strand, which stimulates transcriptional modulation of the expression of target genes [14,15,16]. The WRKYs have been involved in several biological systems in plants, including seed dormancy, seed germination, senescence, and seed development. Similarly, these are also involved in response to biotic and abiotic stresses and switching transcription of their target genes [17,18,19,20].

It is reported that a group of WRKY TFs or one WRKY TF transcriptionally induced by only one stress modulates several stresses. The W-boxes are enriched inside promoter regions [21,22,23,24]. These studies depict the presence of WRKY networks implicated in response to a particular stress, or a combination of two or more. The role of WRKY proteins and underlying mechanisms in response to biotic and abiotic stresses have been frequently investigated in the past decade; however, emphasis of these studies was primarily on one gene in response to one stress in model plants, e.g., *A. thaliana* and *O. sativa*. Recently, an important functional variance among close homologs of WRKY TFs from various plant species has been suggested [24,25]. The functions of WRKYs and WRKY web in non-model plants such as *Capsicum annuum* L., in response to single or combined stresses, has been poorly investigated until now.

Pepper is globally known for its economic importance. It is sown in uplands during temperate seasons [26,27]. Pepper is exposed to various soil-borne diseases such as pepper blight and bacterial wilt caused by bacterial pathogens, i.e., *Phytophthora capsica* and *Ralstonia solanacearum*, respectively [28,29]. The coincidence of these pathogens imparts devastating impacts on pepper production under high temperature and high humidity (HTHH). The HTHH impairs pepper’s immunity mediated by R protein and speeds up pathogen growth and development. Furthermore, pathogen attack and HTHH simultaneously exert natural selection pressure on pepper, which has affected pepper’s evolution in the past [30,31,32]. Conversely, simultaneous occurrence of pathogen infection and HTHH might not affect evolution of model plants, like *A. thaliana* and *O. sativa*. Hence, pepper seems more appropriate for research regarding immunity against biotic or abiotic stresses, i.e., RSI or HTHH.

*Ralstonia solanacearum* causes bacterial wilt in numerous crops, i.e., pepper, tomato, potato, etc. It is a soilborne gram-negative β-proteobacterium [33]. It forms rapid colonies in the xylem tissues, which leads to bacterial wilt disease in the infested plants [34].

The responses of plant species to biotic and abiotic stress are known to be regulated by phytohormones [35]. Phytohormones are tiny molecules involved in plant growth regulation, plant reproduction, and survival. Salicylic acid (SA) and jasmonic acid (JA) are critical for immune responses of plant species. These two are regarded as critical foundations of the immune responses of plants against pathogens. Biosynthesis and signaling of SA is vital for defense against biotrophic pathogens, while JA provides defense against necrotrophic pathogens [36,37]. Ethylene (ET) assists in defense responses against several plant–pathogen interactions [37].

Various indigenous pepper cultivars of subtropical areas demonstrate accelerated disease resistance, even under HTHH [38]. Heat related *cis*-element HSE generally coexist with phytohormones, including SA, JA, ET, ABA, or pathogen related *cis*-elements in promoter regions of most MAPKs and CDPKs involved in plant immunity [39,40,41]. It recommends presence of crosstalk between immunity and HTHH. Pepper genome is 27 and 7.5 times bigger as compared to *A. thaliana* and *O. sativa*, respectively. However, it has only 73 WRKY genes, whereas smaller genomes of *A. thaliana* and *O. sativa* have 72 and 122 WRKY genes, respectively [9,42,43]. We previously detected that *CaWRKY6* [44], *CaWRKY22* [9], *CaWRKY40* [45], and *CaWRKY58* [46] have been involved in pepper’s response against RSI. Among these, *CaWRKY6*, *-22*, *-27*, and *-40* act as positive regulators of plant immunity, whereas *CaWRKY58* is a negative regulator. These genes possess subset of HSE elements and W-boxes in their promoter region and in the promoters of other WRKY TFs, inferring the role of WRKY networks in regulation of pepper’s response against RSI. Additionally, *CaWRKY40* was directly regulated by *CaWRKY6* [44] and *CabZIP63* [16], while *CaWRKY40* was indirectly regulated by *CaCDPK15* [47]. Conversely, most of the pepper’s WRKY TFs have not been characterized yet for its response to pathogen attack.

In the current study, a full-length cDNA for WRKY TFs family of pepper was isolated and named *CaWRKY30*. We characterized it expressionally and functionally and found that *CaWRKY30*was induced by RSI and foliar application of SA, JA, ET, and ABA. Transient over-expression of *CaWRKY30* in pepper leaves triggered HR-like cell death and H_2_O_2_ production. Silencing of *CaWRKY30* in pepper compromised the immunity of pepper to RSI. These results suggest that *CaWRKY30* positively regulates pepper’s immunity to RSI.

## 2. Results

### 2.1. Cloning and Sequence Analysis of CaWRKY30

A new WRKY gene CaWRKY30 (CA01g34480) was identified by genome wide analysis (http://passport.pepper.snu.ac.kr (accessed on 25 October 2021). The *CaWRKY30* gene was chosen for functional characterization, since the existence of immunity-related *cis* elements such as TGA (TGACG motif-binding factor), TGACG-motif, TATC-box, TATA-box, and W-box in the promoter region designate its possible role in pepper’s immunity (Appendix A). A cDNA fragment of *CaWRKY30* (CA01g34480) of 924 bp open-reading-frame (ORF) was cloned by using gene specific primers (Appendix A). The length of deduced amino acid sequence of *CaWRKY30* was 307 amino acids residues, possessing conserved WRKY domain, and it was categorized into group III (Figure 1). The predicted protein has a size and theoretical pI of 34.55 Kda and 7.32, respectively. The *CaWRKY30* shares 88, 60, 68, and 57% amino acid identity with *NsWRKY53*, *CcWRKY53*, *NaWRKY53*, and *StWRKY53*, respectively (Appendix A).

### 2.2. Transcriptional Expression of CaWRKY30 under RSI and Treatment with Various Phytohormones

The presence of immunity-associated *cis*-elements in the promoter region of *CaWRKY30* indicated its potential role in pepper’s immunity RSI. To check the notion, qRT-PCR analysis was carried out to assess the transcriptional levels of *CaWRY30* upon RSI. The transcriptional levels of *CaWRKY30* were increased in *Ralstonia*-treated leaves compared to mock-treated leaves (Figure 2A). The enhanced *CaWRKY30* transcriptional expression levels were consistent between 0 h to 48 h post treatment (hpt). The highest transcriptional expression levels were recorded at 48 hpt (Figure 2A).

Phytohormones, including ET, ABA, SA, and JA mediated signaling pathways, are key regulators of plant reactions to biotic and abiotic stresses. The regulation of *CaWRKY30* by phytohormones-mediated signaling pathways was studied by foliar application of ET, ABA, SA, and JA using qRT-PCR analysis. The results indicated that relative transcriptional expression levels of *CaWRKY30* were enhanced from 0 h to 48 hpt with 100 µM ET as compared to mock. These transcriptional levels were highest at 48 hpt with ET (Figure 2B). Foliar application of 100 µM ABA significantly increased transcriptional levels of *CaWRKY30* as compared to mock. Transcriptional levels reached to the highest level at 12 hpt with ABA (Figure 2C). Results indicated that transcriptional accumulation of *CaWRKY30* was significantly increased after application of 1 mM SA, as compared to mock. These enhanced transcriptional expression levels were highest at 24 hpt (Figure 2D). 

### 2.3. CaWRKY30 Was Found to Be Localized in Nucleus 

Sequence analysis by using WoLFPSORT (“http://www.genscript.com/psort/wolf_psort.html\T1\textquotedblrightolf_psort.htm (accessed on 29 October 2021)) depicted that predicted *CaWRKY30* protein sequence possesses a putative nuclear localization signal (Figure 1), showing its potential targeting in nucleus. To validate this perception, a *CaWRKY30* GFP fusion construct was generated that was driven by constitutive promoter of *CaMV35S*. Afterwards, this constituted vector was transferred into *Agrobacterium* strain GV3101. We transiently over-expressed *CaWRKY30* GFP construct in leaves of *Nicotiana benthamiana* by *A. tumefaciens* injection, and GFP signals were examined by confocal fluorescence microscope. Results depicted that GFP signals of *CaWRKY30*-GFP was present in the nuclei, while GFP of control was detected in various subcellular portions, like cytoplasm and nuclei (Figure 3).

### 2.4. Silencing of CaWRKY30by VIGS Curtailed Pepper’s Immunity to RSI and Downregulated Immunity Associated Marker Genes

The *CaWRKY30* was silenced by virus induced gene silencing (VIGS) to study its role in pepper’s immunity. A total of 50 *CaWRKY30*-unsilenced (TRV:00) and 50 *CaWRKY30*-silenced (TRV:*CaWRKY30*) plants were acquired. Six *CaWRKY30* silenced plants were randomly selected to check their efficiency of gene silencing by root inoculation with cells of compatible virulent *R. solanacearum* strain. Our findings indicated that transcriptional levels of *CaWRKY30* were decreased by ~30% in *Ralstonia*-treated *CaWRKY30* silenced plants as compared to unsilenced plants, validating the silencing of *CaWRKY30* (Figure 4A). The *CaWRKY30* silenced plants showed significantly higher susceptibility to RSI as compared to unsilenced plants. Pepper’s susceptibility to pathogen was increased by showing a rise in *Ralstonia* population growth, marked by higher cfu values in *CaWRKY30* silenced pepper plants as compared with unsilenced plants at 3 d and 5 days post inoculation (dpi) (Figure 4B). Histochemical staining assay was done to detect the H_2_O_2_ production and cell necrosis in *Ralstonia* inoculated *CaWRKY30* silenced (TRV:*CaWRKY30*), and unsilenced (TRV:00) pepper leaves. A dark DAB staining (sign of H_2_O_2_ production) and HR-like cell death indicated by dark trypan blue staining was noticed in unsilenced pepper leaves at 48 hpi (hours post inoculation). Conversely, the concentrations of trypan blue and DAB staining were significantly decreased in *CaWRKY30* silenced pepper leaves (Figure 4C). Electrical conductivity—an indicator of ion leakage—was calculated to study plasma membrane damage and cell death after RSI. The results exhibited that unsilenced plants treated with RSI exhibited higher ion leakage than *Ralstonia*-treated *CaWRKY30* silenced plants at 24 and 48 hpi (Figure 4D). Relative disease index was estimated up to 10 dpi for inferring the level of disease in *CaWRKY30*-silenced and un-silenced plants after RSI (Appendix A). Prominent disease symptoms were noticed in *CaWRKY30* silenced plants at 10 dpi, while unsilenced plants expressed little disease symptoms (Figure 4E). Six *CaWRKY30*-silenced and un-silenced plants were randomly chosen and infiltrated with *R. solanacearum* in the roots for phenotype assay. Obvious wilting disease symptoms were noticed in *CaWRKY30* silenced pepper at 10 dpi, while unsilenced plants expressed very feeble wilting symptoms (Figure 4F). The qRT-PCR analysis was carried out to study the transcriptional accumulation of immunity related marker genes and results expressed that transcriptional expression levels of immunity associated marker genes such as *CaPR1*, *CaNPR1*, *CaDEF1*, *CaABR1,* and *CaHIR1* were reduced in *CaWRKY30*-silenced plants as compared to un-silenced plants at 48 hpi (Figure 4G).

### 2.5. CaWRKY30 Transient Over-Expression Stimulated HR-Like Cell Death, Production of H_2_O_2_, and Transcriptional Upregulation of Immunity Related Marker Genes 

The VIGS experiments indicated that *CaWRKY30* positively regulates pepper’s immunity to RSI. Transient over-expression experiments were performed by infiltrating GV3101 cells carrying 35S:00 (EV) or 35S:*CaWRKY30* into pepper leaves for further validation of this speculation. We studied the role of *CaWRKY30* transient over-expression on induction of HR-like cell death and H_2_O_2_ accumulation in pepper plants. The expression of *CaWRKY30* in pepper plants was established by western blotting experiments (Figure 5A). The HR like cell death and H_2_O_2_ accumulation was confirmed by trypan blue and DAB staining, respectively. Results revealed that transient over-expression of *CaWRKY30* caused HR-like cell death and H_2_O_2_ accumulation in pepper leaves manifested by darker trypan blue staining and dark brown DAB staining (Figure 5B). Conductivity was performed to study the intensity of cell necrosis caused by transient over-expression of *CaWRKY30*, and results indicated that pepper leaves transiently over-expressing *CaWRKY30* showed more ion leakage at 24 and 48 h after Agro-infiltration as compared with pepper leaves over-expressing empty vector (Figure 5C). Relative transcriptional expression levels of defense-associated marker genes such as ET-biosynthesis related *CaPR1*, SA-related *CaNPR1*, ABA-responsive *CaABR1*, and HR-associated marker gene *CaHIR1* were detected by qRT-PCR analysis.

Our results depicted that transient over-expression of *CaWRKY30* in pepper leaves notably amplified transcriptional accumulation of *CaPR1*, *CaNPR1*, *CaDEF1*, *CaABR1*, and *CaHIR1* in pepper as compared to transiently over-expressing empty vector pepper plants at 48 hpi (Figure 5D). These results portray that *CaWRKY30* acts as a positive regulator of plant HR-like cell death, H_2_O_2_ production, and transcriptional upregulation of immunity related marker genes.

### 2.6. Inter-Relationship between CaWRKY30 and CaWRKY40

It is reported that different WRKY TFs form networks together and we previously reported that various members of WRKY TFs, i.e., *CaWRKY6*, *CaWRKY22*, and *CaWRKY40*, are expressionally and functionally linked to each other. Therefore, we postulate that *CaWRKY30* might relate to other members of the WRKY family, including *CaWRKY40*, which is a positive regulator of pepper’s immunity against RSI. The qRT-PCR analysis was performed for further investigation of possible feedback regulation of *CaWRKY40* by *CaWRKY30*, and to study the potential modulation of *CaWRKY30* by *CaWRKY40* transient over-expression assay or by silencing of *CaWRKY40* by VIGS.

Our results expressed transcriptional levels of *CaWRKY30* in pepper leaves transiently over-expressing *CaWRKY40* increased at 24 and 48 hpi in comparison with plants transiently over-expressing empty vector (EV) (Figure 6A), whereas the transcriptional levels of *CaWRKY40* increased in those plants transiently over-expressing *CaWRKY30* as compared to plants transiently over-expressing EV (Figure 6B). Contrastingly, our results depicted that transcriptional accumulation of *CaWRKY40* was notably downregulated in *Ralstonia*-infected *CaWRKY30*-silenced plants as compared to un-silenced plants (Figure 6C). Additionally, transcriptional accumulation of known immunity linked marker genes, e.g., *CaPR1*, *CaNPR1*, *CaDEF1*, *CaABR1,* and *CaHIR1* was fully or partially suppressed in *CaWRKY30* silenced plants that was triggered by *CaWRK40* (Figure 6D).

This data indicates transcriptional regulation of *CaWRKY40* by *CaWRKY30* and the presence of positive regulatory loop involving *CaWRKY30* and *CaWRKY40*.

### 2.7. Inter-Relationship among CaWRY30 and CaWRKY6, CaWRKY22, and CaWRKY27

In our previous studies, *CaWRKY40* was detected to be associated expressionally and functionally to different WRKY TFs, such as *CaWRKY6* and *CaWRKY22*. The relationship among *CaWRKY30* and *CaWRKY40* indicates that *CaWRKY30* is possibly linked with different WRKY TFs involved in pepper’s immunity against RSI. To confirm this assumption, qRT-PCR analysis was performed to study relationship among *CaWRKY30* and different WRKY TFs which have been previously reported to be involved in pepper’s immunity against RSI, including *CaWRKY6*, *CaWRKY22*, and *CaWRKY27*. Our results indicated that transcriptional accumulation of *CaWRKY30* was amplified in *CaWRKY6*, *CaWRKY22,* and *CaWRKY27* transiently over-expressing pepper plants at 48 hpi (Figure 7A), while transcriptional accumulation of *CaWRKY6*, *CaWRKY22*, and *CaWRKY27* were also increased in pepper leaves transiently over-expressing *CaWRKY30* at 48 hpi (Figure 7B). Our results of qRT-PCR assay imply that *CaWRKY30* and *CaWRKY6*, *CaWRKY22*, and *CaWRKY27* are expressionally and functionally inter-related. 

## 3. Discussion

The WRKY protein is one of the largest families of plant transcription factors (TFs). A group of WRKY TFs family members, reported from *Arabidopsis thaliana* and *Oryza sativa*, is known to take part and play a crucial part in plants’ immunity regulation. Considerable functional variations among closely structural homologs of WRKYs have been reported [17,48], and functions of WRKY TFs in immunity of non-model plants, such as pepper, is inadequately investigated.

The *CaWRKY30*, a group III WRKY gene, has been functionally characterized in the current study. Results suggested that *CaWRKY30* positively regulates pepper’s immunity against RSI, and it is a vital constituent of WRKY web comprising *CaWRKY6*, *CaWRKY22*, *CaWRKY27*, and *CaWRKY40*. The entanglement of *CaWRKY30* in pepper’s immunity is endorsed by pathogen responsive *cis* elements, including TGA element, TGACG-motif, TATC-box, TATA-box, and W-box present in the promoter region of *CaWRKY30*. This assumption was consistent with upregulation of transcriptional levels of *CaWRKY30* after RSI, and upon exogenously supplied ETH, ABA, and SA [49,50].

Since genes triggered by exposure to a certain stress factor have been often known to be involved in response to that stress [51], we speculated that *CaWRKY30* might positively regulate pepper’s immunity to RSI. This speculation was supported by the data of silencing of *CaWRKY30* by VIGS, and by *CaWRKY30* transient over-expression assay. This curtailed immunity due to loss of function of *CaWRKY30* was accompanied by amplified growth of RSI and downregulation of immunity related marker genes like SA dependent *CaPR1* [52], *CaNPR1* [53], JA related *CaDEF1* [54], ABA associated *CaABR1* [55], and HR-associated *CaHIR1* [56]. On the other hand, HR-like cell death and H_2_O_2_ secretion was triggered by the transient over-expression of *CaWRKY30* accompanied with transcriptional up regulation of immunity related marker genes like *CaPR1*, *CaNPR1*, *CaDEF1*, *CaABR1,* and *CaHIR1*. This suggests the role of *CaWRKY30* as positive regulator of HR-like cell death and pepper’s immunity against RSI. Likewise, it has been observed in previous studies that *AtWRKY30*, a homolog of *CaWRKY30* in *Arabidopsis*, positively regulates plants’ immunity against biotic and abiotic stresses [57,58]. This can be suggested that *CaWRKY30* is upregulated upon *Ralstonia* infection, leading to decreased vulnerability of pepper to bacterial pathogen infection.

Phytohormones such as ET, ABA, and SA are essential signaling molecules implicated in response of plants to pathogen attack and high-temperature stress. These are also involved in crosstalk between plants’ responses to biotic and abiotic stresses [59]. The SA instigates resistance against biotrophic pathogen, while ET plays a vital role in immunity of plants against necrotrophic pathogens [60]. Generally, synthesis of SA, ET, and JA is coupled with PAMP-triggered immunity (PTI) or effector-triggered-immunity (ETI). These phytohormones can play their role both synergistically and antagonistically based on their concentrations during defense signaling [61]. Synergistic association between these three signaling elements has been detected in PTI, whereas compensatory association has been detected in ETI [62]. The *CaWRKY30* was constantly induced by exogenously supplied phytohormones. The known SA, ET, and ABA-dependent immunity-related marker genes such as *CaPR1*, *CaNPR1*, *CaDEF1*, and *CaABR1* were downregulated by silencing of *CaWRK30*; however, they were upregulated by transient over-expression of *CaWRKY30* in pepper plants, showing that *CaWRKY30* takes part in synergistically mediated defense signaling by SA, ET, and ABA, leading to PTI.

Genome-wide analysis showed the involvement of several WRKY TFs in plant immunity [63,64,65]. By functional genomics studies, *WRKY1* [66], *-11* [67], *-17* [67], *-18* [68], *-22* [69], *-25* [70], *-28* [71], *-33* [72], *-38* [73], *-40* [68], *-45* [74], *-46* [75], *-53* [76], *-54* [75], *-60* [68], *-62* [77], *-70* [75], and *-75* [78] have been functionally characterized in immunity of *Arabidopsis*, regulating immunity positively or negatively. These WRKY TFs have been suggested to arrange themselves into a WRKY transcriptional web, comprising of +ve and –ve feedback loops and feed forward modules [12]. However, configuration of these WRKY networks remains poorly understood. In earlier studies we discovered that *CaWRKY6*, *CaWRKY22*, *CaWRKY27*, and *CaWRKY40* are positive regulator of pepper’s immunity to RSI [9,44,45], while *CaWRKY58* is a negative regulator [46]. Current study infers that *CaWRKY30* acts as positive regulator in pepper’s HR-like cell death, and response of pepper to *R. solanacearum* infection. Transcriptional expression levels of *CaWRKY30* were upregulated upon transient over-expression of *CaWRKY6*, *CaWRKY22*, *CaWRKY27*, and *CaWRKY40*. Conversely, *CaWRKY30* transient over-expression upregulated the transcriptional accumulation of *CaWRKY6*, *CaWRKY22*, *CaWRKY27*, and *CaWRKY40*, suggesting the presence of WRKY TFs networks and +ve feedback loops among *CaWRKY6*, *CaWRKY22*, *CaWRKY27*, and *CaWRKY40*. Alike +ve feedback loops are supposed to be involved in immunity of plants [47]. The same sort of +ve feedback loop has been present among *CaWRKY40* and *CaWRKY6*, *CaWRKY40* and *CabZIP63, CaWRKY40* and *CaCDPK15*. In previous studies, *CaWRKY6* [44] and *CabZIP63* [16] also have been detected to be involved directly and transcriptionally in the regulation of *CaWRKY40* expression against RSI. Keeping in view these results, it can be suggested that *CaWRKY40* might be coordinated by various TFs. In conclusion, *CaWRKY40* expression is co-regulated by *CaWRKY6*, *CaWRKY22*, *CaWRKY30*, and *CabZIP63* upon RSI, and *CaWRKY40* expression is regulated by *CaWRKY6*, and *CabZIP63*, but not by *CaWRKY30* in pepper after exposure to high temperature. 

## 4. Materials and Methods

### 4.1. Plant Materials and Growth Conditions

Seeds of pepper cultivar ‘Mexi’ and *Nicotiana benthamiana* were obtained from Ayub Agriculture Research Institute, Faisalabad, Pakistan. Pepper and *N. benthamiana* seeds were planted in plastic pots having soil mix [peat moss and perlite; 2/1 (*v*/*v*)] and kept in a growth room under controlled conditions, i.e., 25 °C temperature, 60–70 µmol photons m^−2^s^−1^ light intensity, and 70% relative humidity under 16 h light/8 h dark photoperiod. 

### 4.2. Generation of Vectors

The gateway-cloning technique was employed to generate vectors. To construct satellite vectors, *CaWRKY30* full length ORF (with terminal codon or without terminal codon) was cloned into entry vector pDONR207 by using a BP reaction. Afterwards, this construct was transferred into destination vectors pMDC83, CD3687 (HA-tag), CD3688 (Flag-tag), and Pk7WG2 by performing LR reaction to generate vectors for sub-cellular localization and over-expression, respectively.

For the construction of vectors for VIGS, 364 bps fragment in 3′-untranslated region (UTR) of *CaWRKY30* was selected and the specificity was confirmed by BLAST against genome sequence in database of CM334 (http://peppergenome.snu.ac.kr/ (accessed on 29 October 2021)) and Zunla-1 (http://peppersequence.genomics.cn/page/species/blast.jsp (accessed on 29 October 2021)). Next, the BP reaction was performed to clone this fragment into pDONR207/201, after this construct was transferred into vector PYL279 by performing LR reaction. 

### 4.3. Pathogen’s Growth and Inoculation Procedures

We isolated a compatible virulent strain of *R. solanacearum* from pepper plants infected by *Ralstonia* from Dera Ghazi Khan, Pakistan. The tetrazolium chloride method was used to purify above-ground vascular tissue portion exudates of the pathogen-infected plants [9,45]. The isolated *R. solanacearum* was cultivated in SPA medium in thermo control shaker (200 g potato, 20 g sucrose, 3 g beef extract, 5 g tryptone, and 1 L of double-distilled H_2_O) at 200 rpm and 28 °C for overnight. Next, cultivated *R. solanacearum* strain was centrifuged at 6500 rpm and 28 °C for 10 min. Liquid supernatant after centrifugation was poured out, and pellets in the bottom of centrifuge tube were diluted in sterilized 10 mM MgCl_2_ solution. Bacterial cell density was adjusted to 10^8^ cfu ml^−1^ (OD600 = 0.8). To study the effects of RSI on transcriptional levels of *CaWRKY30* and on the resistance of pepper plants to *R. solanacearum* attack, the top third leaf of pepper plants was infiltrated with 10 µL of *R. solanacearum* solution by a syringe without a needle. Pathogen-treated leaves samples were harvested at specific time points for histochemical staining experiments (e.g., DAB staining or trypan blue staining), electrical conductivity, cfu, and RNA extraction for more experiments. For the phenotype experiment of *CaWRKY30*-silenced and unsilenced plants, roots were injured by using a glass rod and infiltrated with compatible virulent *R. solanacearum*. Plants were kept in a growth chamber in control conditions at 28 ± 2 °C temperature, 60–70 µmol photons m^−2^ s^−1^ light intensity, 70% relative humidity, and under a 16-hlight/8-h dark photoperiod after *R. solanacearum* inoculation. Pictures of phenotype were captured after RSI at specific time intervals.

### 4.4. Foliar Application of Phytohormones 

To study the effects of phytohormones, healthy pepper plants were sprayed with 100 µM ETH, 100 µM ABA, and 1 mM SA at four leaf stages. Mock (control) plants were treated with sterile ddH_2_O. Samples treated with phytohormones were collected at desired time points for RNA extraction and for further study. 

### 4.5. Examination of CaWRKY30 Sub-Cellular Localization

*Agrobacterium* cells having 35S:*CaWRKY30*-GFP or 35S:GFP (control) were cultivated overnight in LB medium possessing corresponding antibiotics. Cultured media was centrifuged at 6500 rpm, liquid supernatant was poured out, and the pellet in the bottom was diluted in induction medium (10 mM MES, 10 mM MgCl2, pH 5.7, and 150 µM acetosyringone), and set to OD600 = 0.8. *Agrobacterium* containing 35S*:CaWRKY30*-GFP and 35S:GFP were injected into leaves of *N. benthamiana* by using a syringe without needle. The previously described method was used to do 4,6-diamidino-2-phenylindole staining (DAPI) [9]. At 48 hpt, GFP and DAPI fluorescence signals were observed, and pictures were captured by a laser scanning confocal microscope (TCS SP8, Leica, Solms, Germany), with an excitation wavelength of 488 nm and a 505–530 nm band-pass emission filter.

### 4.6. Histochemical Staining

Histochemical staining (Trypan blue and 3, 3′-diaminobenzidine) were carried out as described previously [9,11,79]. To perform trypan blue staining assay, leaves of pepper were boiled in trypan blue solution (10 mL lactic acid, 10 mL glycerol, 10 mL phenol, 40 mL ethanol, 10 mL ddH_2_O, and 1 mL trypan blue) for 30 min. Afterwards, leaves were kept at room temperature for 8 h, immersed into solution of chloral hydrate (2.5 g of chloral hydrate dissolved in 1 mL of distilled water), and de-stained by boiling for 30 min. Chloral hydrate solution was changed multiple times to reduce the background, and then samples were mounted in 70% glycerol. Leaves were placed in 1 mg/mL of DAB solution for overnight for DAB staining assay. The DAB stained leaves were boiled in lactic acid:glycerol:absolute ethanol [1:1:3 (*v*/*v*/*v*)] and then de-stained in absolute ethanol for overnight [80]. The pictures of trypan blue and DAB-stained leaves were taken by a camera and by a light microscope (Leica, Wetzlar, Germany).

### 4.7. Silencing of CaWRKY30 through Virus-Induced Gene Silencing (VIGS)

Previously described Tobacco Rattle Virus (TRV)-based virus induced gene silencing (VIGS) system was carried out for *CaWRKY30* silencing in pepper plants [47,81,82]. *Agrobacterium* GV3101 containing PYL192, PYL279-*CaWRKY30*, PYL279, and PYL279-*PDS* (OD_600_ = 0.8) constructs were mixed in a 1:1 ratio. This mix was infiltrated into 2-weeks old pepper plants’ cotyledons by a syringe without needle. The infiltrated plants were incubated in a growth chamber for 56 h under dark at 16 °C temperature and 45% relative humidity, and then shifted into a growth room in controlled conditions at 25 ± 2 °C, 60–70 µmol photons m^−2^s^−1^, and relative humidity of 70% under a photoperiod of 16 h light/8 h dark.

### 4.8. CaWRKY30 Transient Over-Expression in Pepper Leaves

*Agrobacterium* cells GV3101 containing *CaWRKY30*-Flag and EV were cultivated in LB medium in a thermo control shaker, having corresponding antibiotics overnight to OD_600_ = 1.0. Afterwards, these *Agrobacterium* GV3101 were centrifuged at 6000 rpm and 28 °C for 10 min. After centrifugation, liquid supernatant was removed and the solid pellet in the bottom was diluted to OD_600_ = 0.8 into induction medium (10 mM MES, pH5.4, 10 mM MgCl_2_, 200 μM acetosyringone). This mixture was infiltrated into leaves by a syringe without needle. These transiently over-expressed leaves were observed for HR-like cell death, or samples were collected to perform DAB and trypan blue staining experiments, and for RNA isolation for more experiments.

### 4.9. RNA Extraction and Quantitative Real Time RT-PCR 

TRIzol reagent method (Invitrogen) was used to extract total RNA from leaves samples of treated pepper plants and from mock seedlings. Afterwards, Prime Script RT-PCR kit (TaKaRa, Dalian, China) was used to reverse transcribe extracted RNA. Real-time qRT-PCR analysis was carried out to study relative transcriptional levels of targeted marker genes. Data processing was done according to previously described methods [45,82], with specific primers (Appendix A), using manufacturer^’^s instructions for Bio-Rad Real time PCR system (Bio-Rad, Foster City, CA, USA) and SYBR premix Ex Taq II system (TaKaRa Perfect Real Time).

### 4.10. Estimation of ELECTRICAL Conductivity

Electrical conductivity (ion leakage) was calculated by using the method described previously with small changes [44,83]. Six leaf discs (4 mm in width) were obtained by using a hole-puncher. The discs were washed three times with sterilized ddH_2_O and incubated instantaneously into 10 mL ddH_2_O. After that, these leaf discs were kept in a gently shaking (60 rpm) shaker at room temperature for 1 h. A conductivity meter (Mettler Toledo 326 Mettler, Zurich, Switzerland) was used to record data of electrical conductivity.

### 4.11. Immuno Blotting 

A previously described procedure of protein extraction buffer was used to extract protein from pepper leaves [3]. The extracted protein was incubated overnight with anti-HA agarose at 4 °C (Thermo Fisher Scientific, Waltham, MA, USA). Magnetic crack was used to collect beads and washed thrice with Tris buffer saline (TBS) and tween 20 (0.05%). Eluted protein was observed using immuno-blotting and by anti-HA-peroxidase (Abcam, Cambridge, UK).

## 5. Conclusions

In conclusion, we are of the view that *CaWRKY30* transcriptionally activates immunity of pepper against RSI. Silencing of *CaWRKY30* by VIGS curtailed pepper plants’ immunity to bacterial pathogen infection, whereas transient over-expression of *CaWRKY30* induced the resistance to *R. solanacearum*. Collectively, the formation of positive feed-back loop and its function in parallel with other TFs is ample evidence of its activation under biotic stress. Additionally, we propose generation of *CaWRKY30* stable transgenic pepper plants and *CaWRKY30* target protein to comprehensively explicate the role of this gene against bacterial pathogen and also in various metabolic pathways.

## Figures and Tables

**Figure 1 ijms-22-12091-f001:**
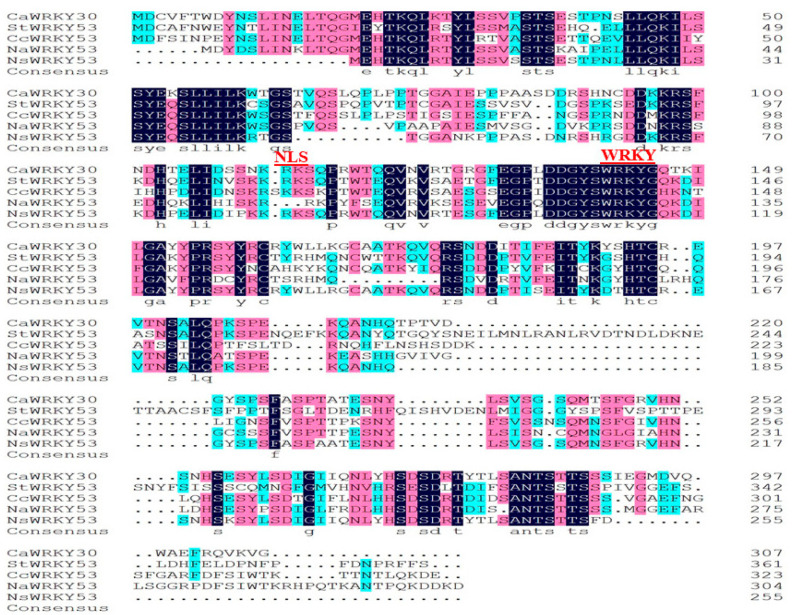
Structural analysis of *CaWRKY30*. Comparison of *CaWRKY30* deduced amino acid sequence with representative related proteins from *Solanum tuberosum StWRKY53* (XP_015166339.1), *Capsicum chinese CcWRKY53* (PHU08488.1), *Nicotiana attenuate NaWRKY53* (XP_019233122.1), and *Nicotiana sylvestris NsWRKY53* (XP_009768977.1). Green shading indicates 50–75% identity, pink shading depicts 75–100% identity, and blue shading indicates 100% identity. Analysis was assayed by using DNAMAN5.

**Figure 2 ijms-22-12091-f002:**
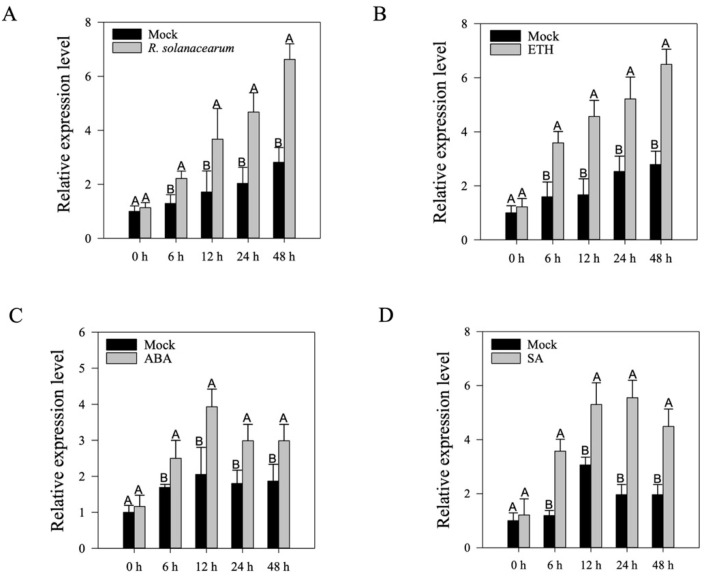
The qRT-PCR assay of *CaWRKY30* relative transcriptional expression levels in pepper leaves subjected to *Ralstonia solanacearum* inoculation (RSI), and foliar application of various phyto-hormones. The qRT-PCR analysis was used to check the transcriptional levels of *CaWRKY30* in pepper leaves subjected to various treatments including RSI (**A**), application of 100 µm ET (**B**), application of 100 µm ABA (**C**), and application of 1 mM SA (**D**) at different time intervals. The transcriptional abundance in RSI-treated leaves was compared with MgCl_2_-treated control leaves (mock), whose relative expression level was set to “1”. (**B**–**D**) The transcriptional expression levels in phyto-hormone-treated leaves were compared with ddH_2_O-treated leaves (mock), whose expression level was set to “1”. The height of the bar indicates means, and error bars indicate the standard error of means. Different letters above the bars show a significant difference between the means based on the Fisher’s protected LSD test.

**Figure 3 ijms-22-12091-f003:**
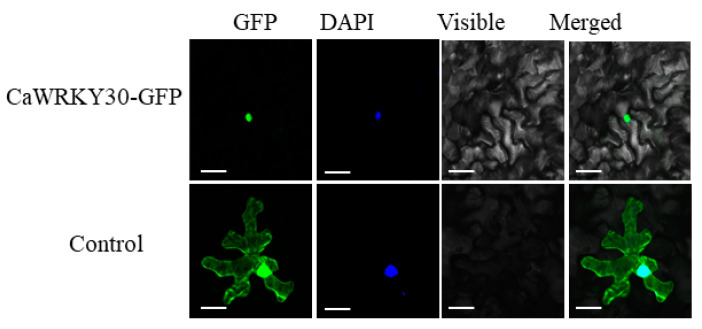
Subcellular localization of *CaWRKY30*. The *CaWRKY30* was fully located in the nucleus of *Nicotiana benthamiana* leaves. Green color demonstrates GFP. Blue color demonstrates DAPI staining of nucleus. Cyan color demonstrates fusion of green GFP and blue DAPI stained nucleus. GFP signal (Green) for the control *N. benthamiana* leaves was detected all over the cell. Images were captured by confocal microscopy at 48 h post inoculation. Bars = 25 µm.

**Figure 4 ijms-22-12091-f004:**
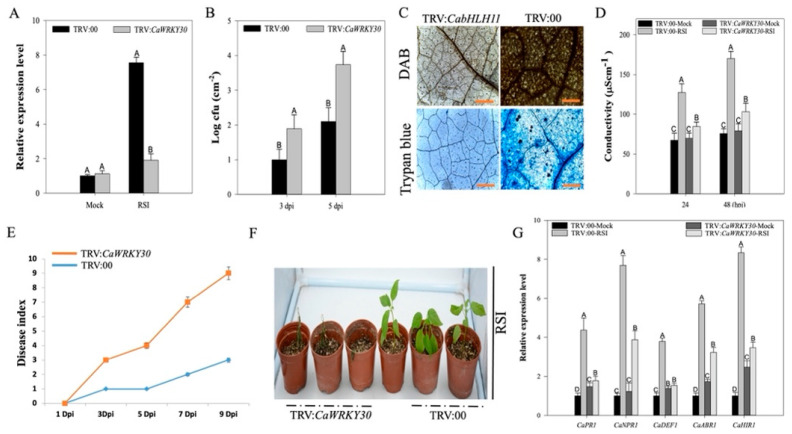
Silencing of *CaWRKY30* by VIGS curtailed pepper’s resistance to RSI and upregulated immunity associated marker genes. The qRT-PCR assay of *CaWRKY30* expressional levels in *R. solanacearum*-inoculated, mock (inoculated with MgCl_2_ solution) *CaWRKY30*-silenced pepper plants (TRV:*CaWRKY30*), and control plants (TRV:00) (**A**). Difference in growth of *R. solanacearum* between *CaWRKY30*-silenced and un-silenced (control) pepper plants inoculated with *R. solanacearum* at 3 and 5 dpi (**B**). Histochemical staining (DAB and trypan blue staining) in *R. solanacearum*-inoculated *CaWRKY30*-silenced (TRV:*CaWRKY30*) and un-silenced (TRV:00) pepper leaves at 48 hpi. Scale bar = 50 µm (**C**). Electrolyte leakage measurement as ion conductivity to evaluate the cell-death responses in the leaf discs of *CaWRKY30*-silenced (TRV:*CaWRKY22*) and un-silenced (TRV:00) pepper plants at 24 and 48 hpi with and without *R. solanacearum* (**D**). *CaWRKY30-*silencing by VIGS enhanced pepper plants susceptibility to *R. solanacearum* infection. Disease index significantly escalated with the passage of time. *CaWRKY30-*silenced (TRV:*CaWRKY30*) plants expressed high susceptibility to *R. solanacearum* infection as compared to un-silenced (TRV:00) plants with the passage of time. Averages are based on four biological replicates with five plants per replication (**E**). Phenotypic effect of *R. solanacearum* treatment on *CaWRKY30*-silenced (TRV:*CaWRKY30*) and un-silenced (TRV:00) pepper plants at 10 dpi (**F**). qRT-PCR assay of transcriptional levels of defense-associated marker genes in *CaWRKY30*-silenced (TRV:*CaWRKY30*) and un-silenced (TRV:00) pepper plants at 48 h post inoculation with *R. solanacearum* (**G**). The relative transcriptional expression level of mock-treated un-silenced plants was set to “1”. The height of the bar indicates means and error bars indicate the standard error of means. Data represents the means ± SE from four biological replicates. Different letters above the bars show significant differences among means.

**Figure 5 ijms-22-12091-f005:**
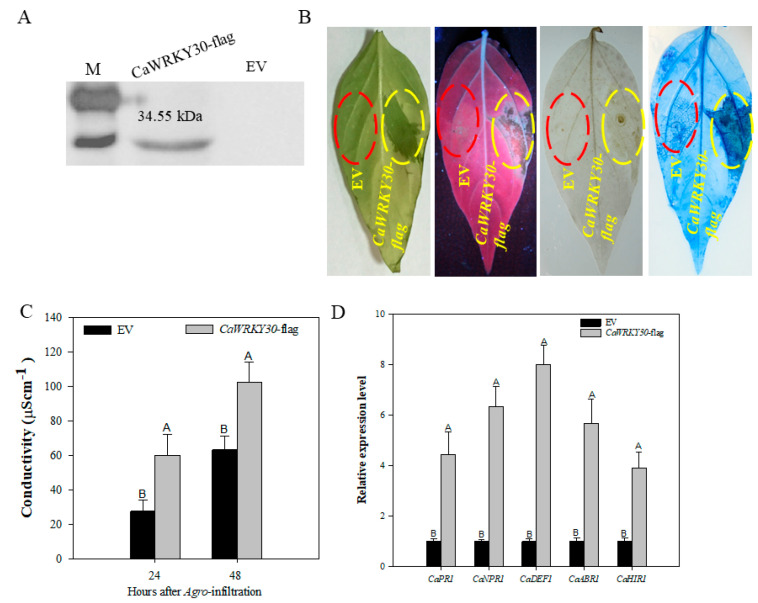
Transient over-expression of *CaWRKY30* induced HR-like cell death, H_2_O_2_ accumulation, and expression of immunity-associated marker genes in pepper leaves. Successful over-expression of *CaWRKY30*-Flag was confirmed by western blotting experiment (**A**), HR-like cell-death triggered by transient over-expression of 35S:*CaWRKY30*, confirmed by phenotype detection, exposure under UV light, and DAB and Trypan Blue staining at 48 hpi, respectively; Scale bar = 50 µm (**B**), Measurement of electrolyte leakage (ion conductivity) to estimate the cell death response in pepper leaf discs at 24 and 48 h after agro-infiltration, respectively (**C**), qRT-PCR assay to check the transcriptional expression levels of immunity-associated marker genes, including *CaPR1*, *CaNPR1*, *CaDEF1*, *CaABR1,* and *CaHIR1* in 35S: *CaWRKY30* expressed pepper leaves at 48 hpi, respectively (**D**). The relative expression level of known defense-related marker-genes in pepper leaves transiently over-expressing the empty vector was set to “1”. Data representing the means ± SE from four biological replicates. Error bars are standard errors of the means. Different letters above the bars indicates significant differences among the means.

**Figure 6 ijms-22-12091-f006:**
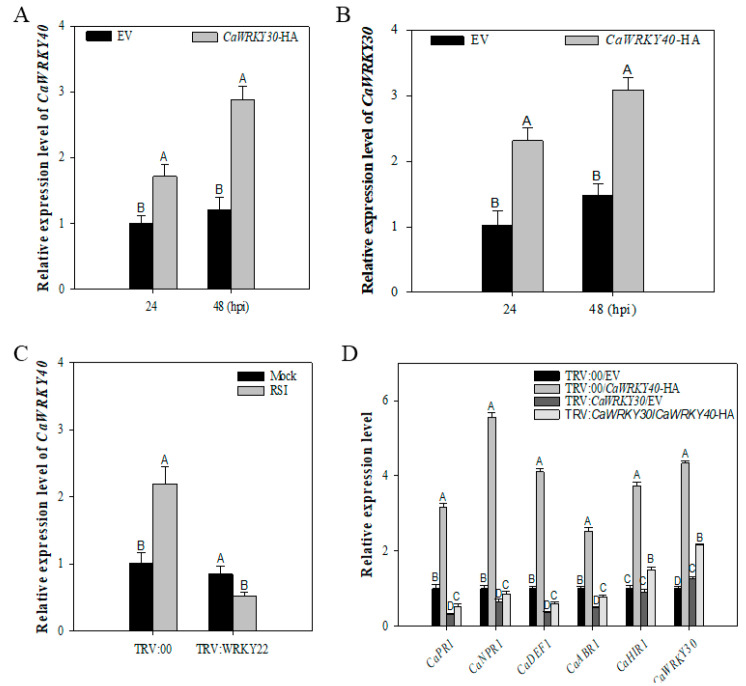
Inter-relationship between *CaWRKY30* and *CaWRKY40*. Transcriptional expression levels of *CaWRKY40* in pepper’s leaves transiently over-expressing *CaWRKY30* at a time interval of 24 and 48 hpi (**A**), Transcriptional expression levels of *CaWRKY30* in pepper’s leaves transiently over-expressing *CaWRKY40* at time interval of 24 and 48 hpi (**B**), qRT-PCR analysis of *CaWRKY40* transcriptional expression levels in*CaWRKY30*-silenced and un-silenced pepper plants at 48 hpi (**C**), qRT-PCR assay of the transcriptional levels of defense-associated marker genes in *CaWRKY30*-silenced and un-silenced pepper plants transiently over-expressing 35S: *CaWRKY40*-HA and 35S:00 at 48 hpi (**D**). The data are means ±SE from four biological replicates. Error bars are standard errors of the means. Different letters above the bars show significant differences among means.

**Figure 7 ijms-22-12091-f007:**
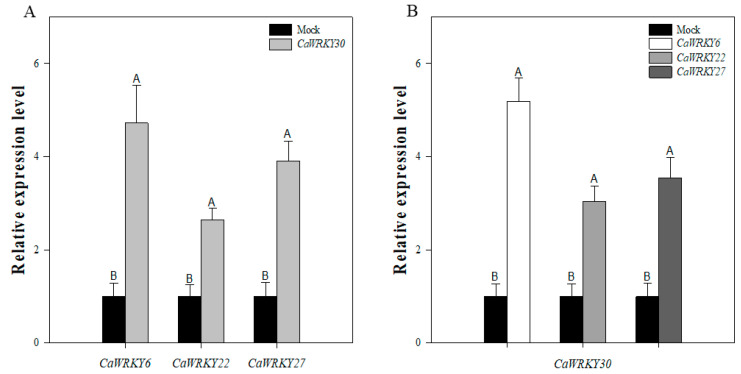
Inter-relationship between *CaWRKY30* and *CaWRKY6*, *CaWRKY22*, and *CaWRKY27*. Transcriptional expression level of *CaWRKY30* in pepper’s leaves transiently over-expressing*CaWRKY6*, *CaWRKY22*, and *CaWRKY27*at 48 hpi (**A**), Transcriptional expression of*CaWRKY6*, *CaWRKY22*, and *CaWRKY27* in pepper’s leaves transiently over-expressing *CaWRKY30* at 48 hpi (**B**). Data are means ± SE from four biological replicates. Error bars are standard errors of the means. Different letters above the bars show significant differences among means.

## Data Availability

All data are within the manuscript and its Appendix A.

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
