# Peer review of "CaWRKY30 Positively Regulates Pepper Immunity by Targeting CaWRKY40 against Ralstonia solanacearum Inoculation through Modulating Defense-Related Genes"

_ijms, 2021, doi:10.3390/ijms222112091_

Round 1

Reviewer 1 Report

ijms-1412099

In this study, CaWRKY30 was characterized functionally in immunity of pepper against Ralstonia solanacearum infection. The CaWRKY30 was detected in nucleus, and its transcriptional expression levels were upregulated by inoculation of Ralstonia solanacearum (RSI), and upon foliar application of plant hormones. Virus induced gene silencing (VIGS) of CaWRKY30 amplified pepper’s vulnerability to RSI. Additionally, the silencing of CaWRKY compromised the HR like cell death triggered by RSI and down regulated defense-associated marker genes. Conversely, transient over expression of CaWRKY30 in leaves of pepper instigated obvious hypersensitive cell death response and up regulated defense related maker genes. Furthermore, transient over expression of CaWRKY30 up regulated transcriptional levels of CaWRKY6, CaWRKY22, CaWRKY27 and CaWRKY40. On the other hand, transient over expression of CaWRKY6, CaWRKY22, CaWRKY27 and CaWRKY40 up regulated the transcriptional expression levels of CaWRKY30. The data collectively recommend that newly characterized CaWRKY30 positively regulates immunity of pepper against bacterial pathogen Ralstonia attack, which is governed by synergistically mediated signaling by phytohormones, and transcriptionally assimilating into WRKY TFs networks. The authors concluded that present data will facilitate to explicate the underlying mechanism of crosstalk between peppers immunity and response to RSI.

This manuscript will contain important findings and be potentially interesting for readers. I have some opinions to the manuscript.

  • The authors newly isolated R. solanacearum virulent to pepper cultivar Mexi, and used throughout experiments. In figure 4, authors estimated both disease development and hypersensitive response. I wounder why virulent strain of R. solanacearum caused hypersensitive response. I believe the authors should use incompatible R. solanacearum strain to analyzed hypersensitive response.

  • In figure 2. The authors estimated CaWRKY30 expression by treatment with phytohoromones. I wounder why CaWRKY30 induced Synchronously by the treatment with JA and SA. Are there no competitive relationship between JA and SA on CaWRKY30 expression ? The authors should discuss in the text.

  • In figure 4E, I believe statistical analysis is needed. In figure 4F, all TRV00 plants wilted, but all TRV:CaWRKY30 plants still alive. Is it true ?

  • The authors should use clear photo of DAB staining experiment in figure 5.
  • I feel the authors confused hypersensitive response-mediated disease resistance and basal disease resistance. Much of hypersensitive response-mediated disease resistance was induced by incompatible pathogens, effectors and/or elicitors. Conversely, basal disease resistance was induced by compatible and virulent pathogens. Such concept was proposed by Jones and Dangl (2006) in Nature 16;444(7117):323-9.doi: 10.1038/nature05286. The authors should carefully distinct the phenomenon, construct experiments and carefully discuss by including PAMPs-triggered immunity and effector-triggered immunity.

Author Response

REVIEWER 1

In this study, CaWRKY30 was characterized functionally in immunity of pepper against Ralstonia solanacearum infection. The CaWRKY30 was detected in nucleus, and its transcriptional expression levels were upregulated by inoculation of Ralstonia solanacearum (RSI), and upon foliar application of plant hormones. Virus induced gene silencing (VIGS) of CaWRKY30 amplified pepper’s vulnerability to RSI. Additionally, the silencing of CaWRKY compromised the HR like cell death triggered by RSI and down regulated defense-associated marker genes. Conversely, transient over expression of CaWRKY30 in leaves of pepper instigated obvious hypersensitive cell death response and up regulated defense related maker genes. Furthermore, transient over expression of CaWRKY30 up regulated transcriptional levels of CaWRKY6CaWRKY22, CaWRKY27 and CaWRKY40. On the other hand, transient over expression of CaWRKY6CaWRKY22CaWRKY27 and CaWRKY40 up regulated the transcriptional expression levels of CaWRKY30. The data collectively recommend that newly characterized CaWRKY30 positively regulates immunity of pepper against bacterial pathogen Ralstonia attack, which is governed by synergistically mediated signaling by phytohormones, and transcriptionally assimilating into WRKY TFs networks. The authors concluded that present data will facilitate to explicate the underlying mechanism of crosstalk between peppers immunity and response to RSI.

This manuscript will contain important findings and be potentially interesting for readers. I have some opinions to the manuscript.

Response: Thank you very much for the constructive comments. We have incorporated the opinions.

The authors newly isolated R. solanacearum virulent to pepper cultivar Mexi, and used throughout experiments. In figure 4, authors estimated both disease development and hypersensitive response. I wounder why virulent strain of R. solanacearum caused hypersensitive response. I believe the authors should use incompatible R. solanacearum strain to analyzed hypersensitive response.

Response: We examined HR-like cell death, not hypersensitive response. This typo has been corrected

In figure 2. The authors estimated CaWRKY30 expression by treatment with phytohoromones. I wounder why CaWRKY30 induced Synchronously by the treatment with JA and SA. Are there no competitive relationship between JA and SA on CaWRKY30 expression ? The authors should discuss in the text.

Response: The JA was not used in the study. It has been corrected throughout the manuscript.

In figure 4E, I believe statistical analysis is needed. In figure 4F, all TRV00 plants wilted, but all TRV:CaWRKY30 plants still alive. Is it true ?

Response: It was a typo. We have corrected it

The authors should use clear photo of DAB staining experiment in figure 5.

Response: A clear photo of DAB staining has been added

I feel the authors confused hypersensitive response-mediated disease resistance and basal disease resistance. Much of hypersensitive response-mediated disease resistance was induced by incompatible pathogens, effectors and/or elicitors. Conversely, basal disease resistance was induced by compatible and virulent pathogens. Such concept was proposed by Jones and Dangl (2006) in Nature 16;444(7117):323-9.doi: 10.1038/nature05286. The authors should carefully distinct the phenomenon, construct experiments and carefully discuss by including PAMPs-triggered immunity and effector-triggered immunity.

Response: We examined HR-like cell death, not hypersensitive response. This typo has been corrected

Reviewer 2 Report

The research paper entitled ‘CaWRKY30 Positively Regulate Pepper Immunity by Targeting CaWRKY40 against Ralstonia solanacearum Inoculation through Modulating Defense-related Genes by Hussain et al. describes a novel transcription regulator from WRKY group in pepper  and reports its functional and expressional characterization. It is worth to emphasize the reported role of this new TF in setting up an infection by an economically important plant pathogen Ralstonia solanacearum and during exogenous supplementation with phytohormones. Besides, interplay between other TFs from WRKY group is described. The whole topic and the collected data are of high novelty and significance, being definitely worth releasing in International Journal of Molecular Sciences. Though, the manuscript has several notable drawbacks, which are listed below.  

- It would be beneficial to improve the introduction section. There is no information on Ralstonia solanacearum, the utilized phytohormones nor the studied immunity-related genes. Also there are parts (e.g. lines 77-87), which are in my opinion repetitive, lack significant information and are long-winded. Likewise, underlining superiority of the used model plant pepper in contrast to rice and A. thaliana was repeated several times.

- There are editorial issues: e.g. lack of/additional spaces (e.g. lines 91, 110, 122, 125, 171, 178, 207, 244, 252, 273, 375, 491, 562), typos (e.g. lines 32, 210, 397, 520, 548, 549, ), lack of subscripts (line 290)  etc.

- Although I am not a native English speaker myself, English in this manuscript needs to be notably improved. Some examples: lines 77, 106-107, 316-318, 146, 153, 197, 353, 355, 371-373, 378, 385, 395,  399, 435, 477, 520, 527, 532.

- Line 24 –> two corresponding authors are mentioned, but there is only one email address.

- Line 31 -> I am confused whether ethephon (line 31, 172, 384, 402, 406, 412, 486) or ethylene (line 43, 105, 122, 163, 164, 405, 412) was studied

- Line 43 –> these abbreviations have been introduced before

- Results -> the collected results are interesting and in my opinion result from a correctly performed experimental design. However, there are some issues to be addressed in this section (see below).

- Please improve resolution of the included figures – e.g. the axis descriptions look foggy in print

- line 129 -> was identified for the first time by genome wide analysis (http://passport.pepper.snu.ac.kr) which was not characterized previously.

- Fig S1 -> The description of the figure is incorrect. As far as I am aware the authors studied CaWRKY30 not 22. Please assure uniformity between the legend and description e.g. W-Box vs. Wbox. Please be sure that the order of the described motifs follows the ordering in the legend.

- Fig. 1 -> Blue shading, pink shading, not green nor red, respectively (at least at my computer screen)

- Fig 2. -> Next to the information about whiskers with errors, please state to what refers the bar height (probably to means). There are no lowercase letters – this information is unnecessary

- Fig. 4 -> There is no (A) in the figure description. There is something wrong with interpretation of panel F. “…while unsilenced pepper plants expressed very feeble wilting disease symptoms” (line 237) -> according to the photo silenced plants seem fine and the unsilenced wilted. Also panel D suggest downregulation of immunity-related genes not upregulation as stated in line 244. Please decide whether error bars mark SD or SE (line 263). There are no lowercase letters – this information is unnecessary

- Table 3 is introduced in the main text before Table 2

- Fig. 5 -> There is no (A) in the figure description. Please decide whether error bars mark SD or SE (line 303). There are no lowercase letters – this information is unnecessary

- Lines 316-320 -> Please improve this part. I am not sure whether this part suffers from English-related issues or the results from panels AB are incorrectly described.

- Fig. 6 -> Panel C – WRKY30 I image not 22. Please improve description of panel D to make this part easier to interpret. There is no (A) in the figure description. Please decide whether error bars mark SD or SE (line 340, 342). There are no lowercase letters – this information is unnecessary

- Fig. 7 -> Please decide whether error bars mark SD or SE (line 366-367). There are no lowercase letters – this information is unnecessary

- The Discussion section needs substantial improvement of English. This whole section should be rewritten. Too much attention is attributed to repetition of the results, too little to actual discussion. Please explain PTI and ETI during first appearance (407).

- Lines 453, 455 -> please explain abbreviations BP and LR during first appearance

- Line 467 -> media components should be listed after the medium not shaker.

- Materials and Methods section is concise and at the same time provides enough details. Though, it suffers from language issues.

- Supplementary tables: Table 1 -> Pepper-specific primers used for construction of the vectors; Table 2 -> to the sequence of 3’UTR; Table 3 -> apical region, did you mean “top of the pepper plant”?

Author Response

REVIEWER 2

The research paper entitled ‘CaWRKY30 Positively Regulate Pepper Immunity by Targeting CaWRKY40 against Ralstonia solanacearum Inoculation through Modulating Defense-related Genes by Hussain et al. describes a novel transcription regulator from WRKY group in pepper  and reports its functional and expressional characterization. It is worth to emphasize the reported role of this new TF in setting up an infection by an economically important plant pathogen Ralstonia solanacearum and during exogenous supplementation with phytohormones. Besides, interplay between other TFs from WRKY group is described. The whole topic and the collected data are of high novelty and significance, being definitely worth releasing in International Journal of Molecular Sciences. Though, the manuscript has several notable drawbacks, which are listed below.  

Response: Thank you for your constructive criticism. We have improved the manuscript according to the suggestions made by you.

- It would be beneficial to improve the introduction section. There is no information on Ralstonia solanacearum, the utilized phytohormones nor the studied immunity-related genes. Also there are parts (e.g. lines 77-87), which are in my opinion repetitive, lack significant information and are long-winded. Likewise, underlining superiority of the used model plant pepper in contrast to rice and A. thaliana was repeated several times.

Response: Thank you for suggestion. Introduction section is improved. Information on Ralstonia solanacearum, the utilized phytohormones nor the studied immunity-related genes is incorporated. Also lines 77-87, sentenced rephrased and made more comprehensive.

- There are editorial issues: e.g. lack of/additional spaces (e.g. lines 91, 110, 122, 125, 171, 178, 207, 244, 252, 273, 375, 491, 562), typos (e.g. lines 32, 210, 397, 520, 548, 549, ), lack of subscripts (line 290)  etc.

Response: The manuscript has been edited thoroughly and all such mistakes have been removed.

- Although I am not a native English speaker myself, English in this manuscript needs to be notably improved. Some examples: lines 77, 106-107, 316-318, 146, 153, 197, 353, 355, 371-373, 378, 385, 395,  399, 435, 477, 520, 527, 532.

Response: The manuscript has been edited thoroughly for language and grammar useage.

- Line 24 –> two corresponding authors are mentioned, but there is only one email address.

Response: Email addresses of both authors have been added now.

- Line 31 -> I am confused whether ethephon (line 31, 172, 384, 402, 406, 412, 486) or ethylene (line 43, 105, 122, 163, 164, 405, 412) was studied

Response: Ethylene was studied. It has been corrected throughout the manuscript.

- Line 43 –> these abbreviations have been introduced before

Response: Each abbreviation has been expanded at its first cite.

- Results -> the collected results are interesting and in my opinion result from a correctly performed experimental design. However, there are some issues to be addressed in this section (see below).

- Please improve resolution of the included figures – e.g. the axis descriptions look foggy in print,

Response: The resolution of the images have been improved.

- line 129 -> was identified for the first time by genome wide analysis (http://passport.pepper.snu.ac.kr) which was not characterized previously.

Response: Corrected

- Fig S1 -> The description of the figure is incorrect. As far as I am aware the authors studied CaWRKY30 not 22. Please assure uniformity between the legend and description e.g. W-Box vs. Wbox. Please be sure that the order of the described motifs follows the ordering in the legend.

Response: Corrected

- Fig. 1 -> Blue shading, pink shading, not green nor red, respectively (at least at my computer screen)

Response: Corrected

- Fig 2. -> Next to the information about whiskers with errors, please state to what refers the bar height (probably to means). There are no lowercase letters – this information is unnecessary

Response: Corrected

- Fig. 4 -> There is no (A) in the figure description. There is something wrong with interpretation of panel F. “…while unsilenced pepper plants expressed very feeble wilting disease symptoms” (line 237) -> according to the photo silenced plants seem fine and the unsilenced wilted. Also panel D suggest downregulation of immunity-related genes not upregulation as stated in line 244. Please decide whether error bars mark SD or SE (line 263). There are no lowercase letters – this information is unnecessary

Response: The footnotes of all figures have been corrected keeping in view these suggestions

- Table 3 is introduced in the main text before Table 2

Response: Corrected

- Fig. 5 -> There is no (A) in the figure description. Please decide whether error bars mark SD or SE (line 303). There are no lowercase letters – this information is unnecessary

Response: Corrected

- Lines 316-320 -> Please improve this part. I am not sure whether this part suffers from English-related issues or the results from panels AB are incorrectly described.

Response: Corrected

- Fig. 6 -> Panel C – WRKY30 I image not 22. Please improve description of panel D to make this part easier to interpret. There is no (A) in the figure description. Please decide whether error bars mark SD or SE (line 340, 342). There are no lowercase letters – this information is unnecessary

Response: Corrected

- Fig. 7 -> Please decide whether error bars mark SD or SE (line 366-367). There are no lowercase letters – this information is unnecessary

Response: These are SE and corrected throughout the manuscript

- The Discussion section needs substantial improvement of English. This whole section should be rewritten. Too much attention is attributed to repetition of the results, too little to actual discussion. Please explain PTI and ETI during first appearance (407).

Response: Language of the whole manuscript has been improved and all abbreviations have been expanded at their first mentions.

- Lines 453, 455 -> please explain abbreviations BP and LR during first appearance

Response: These are well known reactions and written as such.

- Line 467 -> media components should be listed after the medium not shaker.

Response: Corrected

- Materials and Methods section is concise and at the same time provides enough details. Though, it suffers from language issues.

Response: Thank you! Language has been polished.

- Supplementary tables: Table 1 -> Pepper-specific primers used for construction of the vectors; Table 2 -> to the sequence of 3’UTR; Table 3 -> apical region, did you mean “top of the pepper plant”?

Response: Corrected

Reviewer 3 Report

The article "CaWRKY30 Positively Regulate Pepper Immunity by Targeting 2 CaWRKY40 against Ralstonia solanacearum Inoculation 3 through Modulating Defense-related Genes" seems a good piece of work with well-designed experiment.  However extensive revision in the language is needed for clarity and better presentation.

Here is the some suggestions:

The author can present the matters present between line-64-69 in  a diagram, that will be easy  for reader understanding .

Line-78-78  "It is reported that a group of WRKY TFs transcriptionally induced by only one stress  or one WRKY TF modulates various stresses " sentence is slightly confusing, check it again

Line 89- "Being Solanaceae member"- it is not necessary that being solanaceae member, have high risk of infection or pathogenic attack

Line-92- 94 "The coincidence of these soil-borne pathogens imparts devastating impacts on pepper  production under stress conditions of high temperature and high humidity (HTHH), and this impairs pepper’s immunity mediated by R protein and speeds-up pathogen’ growth  and development" – Reframe the sentence, for clarity

Line 97-101- reframe the sentence

Line 402-404 reframe the sentence

Author Response

REVIEWER 3

The article "CaWRKY30 Positively Regulate Pepper Immunity by Targeting 2 CaWRKY40 against Ralstonia solanacearum Inoculation 3 through Modulating Defense-related Genes" seems a good piece of work with well-designed experiment.  However extensive revision in the language is needed for clarity and better presentation.

Here is the some suggestions:

The author can present the matters present between line-64-69 in  a diagram, that will be easy  for reader understanding .

Response: The lines have been made more clear.

Line-78-78  "It is reported that a group of WRKY TFs transcriptionally induced by only one stress  or one WRKY TF modulates various stresses " sentence is slightly confusing, check it again

Response: The language of the whole manuscript has been polished.

Line 89- "Being Solanaceae member"- it is not necessary that being solanaceae member, have high risk of infection or pathogenic attack

Response: Deleted

Line-92- 94 "The coincidence of these soil-borne pathogens imparts devastating impacts on pepper  production under stress conditions of high temperature and high humidity (HTHH), and this impairs pepper’s immunity mediated by R protein and speeds-up pathogen’ growth  and development" – Reframe the sentence, for clarity

Response: The sentence has been rephrased

Line 97-101- reframe the sentence

Response: The sentence has been rephrased

Line 402-404 reframe the sentence

Response: The sentence has been rephrased

Round 2

Reviewer 1 Report

The authors addressed to my comments and suggestions.